# A Rare Case of Epithelioid Trophoblastic Tumor Presenting as Hematoma of a Caesarean Scar in the Lower Uterine Segment

**DOI:** 10.3390/medicina58010034

**Published:** 2021-12-25

**Authors:** Radomir Aničić, Aleksandar Rakić, Rastko Maglić, Dragutin Sretenović, Aleksandar Ristić, Elena Đaković, Lazar Nejković

**Affiliations:** 1The Obstetrics and Gynecology Clinic Narodni Front, Kraljice Natalije 62, 11000 Belgrade, Serbia; a.r.rakic@gmail.com (A.R.); rastko.maglic@gmail.com (R.M.); mr.gutasret@gmail.com (D.S.); ristic.aleksandar@icloud.com (A.R.); djakovicelena7@gmail.com (E.Đ.); lnejkovic@sbb.rs (L.N.); 2School of Medicine, University of Belgrade, Dr Subotića Starijeg 8, 11000 Belgrade, Serbia

**Keywords:** epithelioid trophoblastic tumor, Caesarean scar, gestational trophoblastic neoplasia

## Abstract

Epitheliod trophoblastic tumor (ETT) account for only 1–2% of all the cases of gestational trophoblastic neoplasia (GTN), with a reported mortality rate of 10–24%. ETT is derived from chorionic type intermediate trophoblastic cells, which appears to be the reason for the only slightly elevated βhCG levels in these patients. We present a case of a 42-year-old patient who was admitted to the clinic eight months after Caesarean delivery, for irregular vaginal bleed with normal values of beta-human chorionic gonadotropin (βhCG). A 6 × 5 cm hematoma was evacuated from the isthmic uterine segment during the operation, and the histopathological exam of the tissue surrounding the hematoma revealed ETT. There were no metastatic lesions on the thoracal, abdominal, and pelvic CT. The second ultrasonographic exam revealed tumefaction of 5 cm at the site from the previous surgical procedure. Color Doppler imaging revealed no central nor peripheral blood flow. The patient underwent a total abdominal hysterectomy with bilateral adnexectomy without adjuvant chemotherapy. This appears to be one of the shortest intervals from the anteceded gestational event until the diagnosis of this tumor, along with the absence of the significant ultrasonographic feature of the ETT-peripheral Doppler signal pattern. We underline that, even with normal values of βhCG, irregular vaginal bleeding following the antecedent gestational event should always arouse suspicion of GTN.

## 1. Introduction

A little less than two decades ago, the World Health Organization (WHO) recognized epithelioid trophoblastic tumor (ETT) as a form of gestational trophoblastic neoplasia (GTN) [1]. First described in 1998 [2], ETT accounts for only 1–2% of all cases of GTN, with a reported mortality rate of 10–24% [3,4]. ETT is derived from chorionic type intermediate trophoblastic cells, forming nests and solid masses with nodular and expansive growths [5]. So far, a little more than 130 cases of ETT have been reported [3], and our knowledge regarding this type of GTN is mostly based on the mentioned isolated cases or small case series. Because of the rareness of the disease and its variable presentation, the gynecological community currently relies on basic guidelines for the management of ETT, and it is far from the universally accepted protocol for diagnosis and treatment. Here, we present a case of ETT mimicking the hematoma on the Caesarean section scar in the lower uterine segment.

## 2. Case Report

A 42-year-old patient, para 2, gravida 2, was admitted to the gynecology department for vaginal bleeding, eight months after Caesarean section delivery. She also complained about irregular bleeding that started two months after delivery. Her serum beta-human chorionic gonadotropin (βhCG) was not elevated. A gynecological exam revealed painful tumefaction around the lower uterine segment. A 5 cm tumefaction in the Caesarean scar region was observed on the ultrasonographic exam. During the explorative laparotomy, a 6 × 5 cm hematoma was evacuated from the isthmic uterine segment. The surrounding pelvic structures appeared to be morphologically intact. A small tissue sample around the hematoma was sent to the histopathological exam and the uterine wall was reconstructed in the two layers. Histopathological exam showed fragments of endocervical tissue with intermediate trophoblastic cells forming elongated structures and nests (Figure 1a,b). The mean mitotic count was two mitoses per 10 high-power fields. Immunohistochemically, the samples showed diffuse strong positivity for CK AE1/AE3 (Figure 2a), Cyclin E (Figure 2b), and p63 (Figure 3a). There was a focal HPL and inhibin positivity. The Ki-67 nuclear labeling index was 20% (Figure 3b). Based on morphological and immunohistochemical findings, the patient was diagnosed with an epithelioid trophoblastic tumor. There were no metastatic lesions on the thoracal, abdominal, and pelvic CT. Serum human chorionic gonadotropin was negative. A tumefaction of 5 cm at the site from the previous surgical procedure was detected ultrasonographically (Figure 4a) during the patient’s second admission to the clinic, a month after the prior hospitalization. The mentioned tumefaction had sharp borders to the surrounding tissue, with an apparent hypoechogenic halo. Color Doppler imaging revealed no central nor peripheral blood flow. Intra-operatively, there was a 5 cm dehiscence at the scar from the previous operation (Figure 4b) with coagulated blood. There was no apparent involvement of the organs available for inspection and palpation. The procedure was uneventful, and the patient was discharged without adjuvant chemotherapy. There are no signs of metastatic disease ten months after the surgery, and the serum human chorionic gonadotropin, as well as human placental lactogen (hPL), remained within normal values.

## 3. Discussion

As with all unfamiliar diseases, our knowledge about ETT is based on case reports, case series, and literature reviews. The variety of the presentation, patient’s age, and different medical histories make the diagnosis of this mysterious disease even more difficult. As of this moment, there are no universally accepted protocols for the diagnostic route(s), treatment regimens, and the duration of the treatment for patients diagnosed with ETT.

We found only a few reports of ETT presenting as a mass in the Caesarean scar, with two being synchronous with choriocarcinoma [6,7,8,9,10]. Black et al. reported a similar case where the patient with three previous Caesarean sections had been diagnosed with ETT after being followed for spontaneous abortion with retained products of conception [6]. This patient presented with transfusion-requiring vaginal bleeding, two months after suction evacuation for retained conception products, along with negative tumor markers [6]. Our patient had a mild vaginal discharge and irregular bleeding of minimal intensity. The main difference between the two cases was that our patient had hematoma and scar dehiscence immediately after Cesarean delivery.

Most ETTs occur in reproductive-age women, but there are also reports of this disease in postmenopausal women [11]. Based on the current literature findings, an average period from, most frequently, term pregnancy, followed by molar pregnancy or abortion, until the diagnosis of ETT, is 76 months [4]. Moreover, the main complaint of the patients with ETT was abnormal uterine bleeding. Therefore, as the International Society for the Study of Trophoblastic Diseases suggested, gestational trophoblastic disease or neoplasia should always be considered in patients with abnormal bleeding after the anteceded gestational event [12].

What is remarkable in this case is that the patient’s mainly used tumor markers for gestational trophoblastic neoplasia returned negative. Serum βhCG, the most commonly used diagnostic tool for gestational trophoblastic neoplasia, is often only moderately elevated in the cases of ETT, with a range of 665–2500 mIU/mL [4]. One of the most recent reports of ETT in a 44-year old patient summarized only eight cases of uterine ETT with a normal values of serum βhCG [13]. Among these cases, the interval to antecedent pregnancy ranged from 1 to 30 years [13], significantly longer than an interval of eight months in our presentation. The eight month interval from the preceding pregnancy until the diagnosis of ETT in our case appears the be one of the shortest reported periods between the gestational event and the tumor diagnosis.

The main explanation for normal levels and moderate elevation in βhCG concentration in patients with ETTs is the origin of these tumors. They arise from intermediate-type chorionic trophoblasts, which produce little βhCG. Although these cell populations compose large quantities of hPL, experience from published case series determines that this is not a valuable tumor marker for ETT. On the other hand, hPL could be a useful marker for the diagnosis of the other rare form of GTN-placental site trophoblastic tumor (PSTT), even though its levels did not reflect disease activity and prognosis [14].

In a retrospective series by Quin et al., all 12 cases of ETT presented as a well-circumscribed border with the hypoechogenic halo on the ultrasonographic exam, which appears the be the distinctive feature of these tumors [15]. In the same series, 11 of the 12 ETT cases had a specific peripheral Doppler signal, rather than the non-peripheral Doppler pattern observed in all cases of PSTT. This series concluded that the peripheral patter of Doppler signal in ETTs could be a discriminative feature from other GTNs [15]. Our patient indeed had tumefaction of sharp borders to the surrounding tissue, with the mentioned hypoechogenic halo. On the contrary, color Doppler imaging revealed no central nor peripheral blood flow. A possible explanation of the lack of a specific peripheral Doppler pattern could be the presentation of the tumor as the tissue around the uterine Caesarean scar dehiscence, not like the usual well-circumscribed solid mass.

A histopathological exam is crucial for the definitive diagnosis of ETT. The exam should be performed by a pathologist with experience in GTN in order to minimize the risk of misdiagnosis, as it is often perplexing to differentiate ETT from other forms of GTN [10]. ETT is usually observed as a solid, well-circumcised lesion in the cervix or as a distant, extra-uterine localized mass [4,16]. The cervical localization of most ETTs is one of the differences between these tumors and uterine corpus located PSTTs [4]. Distinctive of PSTTs, the histological image of ETT shows well-circumscribed, nested growth of smaller cells with less nuclear pleomorphism, often presented with a geographical necrosis pattern. As there are reports of ETT coexisting with PSTT [7], it is necessary to distinguish these two entities histologically. Furthermore, postchemotherapy examined samples of other forms of GTN can present with a histological pattern similar to ETT [17]. These samples have p63 positivity, but a low Ki67 proliferative index [17]. The mitotic count observed in ETT ranges from 0 to 9 per 10 HPFs, and the Ki-67 ranges from 3 to more than 70%. [17]. Further histologic diagnosis is provided with immunohistochemistry. ETTs express diffuse positivity for cytokeratin AE1/AE3, cytokeratin 18, inhibin-α, E-cadherin, Cyclin-E, epithelial membrane antigen (EMA), p63, and prolyl 4-hydroxylase, while the expression of hCG, hPL, and Mel-CAM is focal and usually weak [4,14,16,17,18]. Immunohistochemistry is also a crucial tool for differentiating ETT from other GTNs, as well as squamous cell cervical carcinoma. p63, hPL, and Mel-CAM positivity of ETT cells is useful for distinguishing ETT from PSTT, while inhibin-α and cytokeratin 18 are usually not expressed in squamous cell carcinoma [4,16]. Moreover, ETTs lack the intercellular bridges between the cells, which are present in squamous cell carcinoma [4].

The current opinion is that the primary treatment modality for patients with ETT confined to the uterus is surgical. Total abdominal hysterectomy with bilateral adnexectomy helped the definitive management of the disease in the reported case series. Even though there are some debates, most authors do not recommend adjuvant chemotherapy for these patients. One of the largest case series of patients with ETT underlined that stage I disease without adverse factors (i.e., antecedent pregnancy > 48 months) should undergo surgery alone, and that adjuvant chemotherapy is advised if these factors are present. In their review, Gadducci et al. recommend hysterectomy with bilateral salpingectomy in stage I patients who wish to preserve fertility [4]. They also state that oophorectomy could be avoided in patients with macroscopically normal ovaries [4]. Reports of fertility-sparing treatment methods in patients with ETT are scarce, with the safety and success of these methods being far from proven. Treatment of patients with distant metastases is even more complex and variable than the disease itself and, requires a multimodality approach, including various regimens and durations of chemotherapy, as well as complex surgical procedures.

Our knowledge regarding prognostic factors in patients with ETT relies on larger case series and literature reviews. Some authors believe that risk factors for an unfavorable prognosis are similar to those of PSTT, regardless of the rarity of this disease. So far, the period from the antecedent gestational event longer than 48 months and advanced-stage disease remain the only accepted factors for the poor prognosis in these patients.

## 4. Conclusions

We presented the case of ETT diagnosed eight months following the Cesarean delivery, with what appears to be one of the shortest intervals from the anteceded gestational event until the diagnosis of this tumor. The other distinctive feature of this case was the absence of the significant ultrasonographic feature of the ETT-peripheral Doppler signal pattern. This second reported case of an accidental finding of ETT mimicking hematoma in Caesarean scar in the lower uterine segment further expands the spectrum of ETT’s clinical presentations. We underline that, even with normal values of βhCG, irregular vaginal bleeding following the antecedent gestational event should always arouse suspicion of GTN.

## Figures and Tables

**Figure 1 medicina-58-00034-f001:**
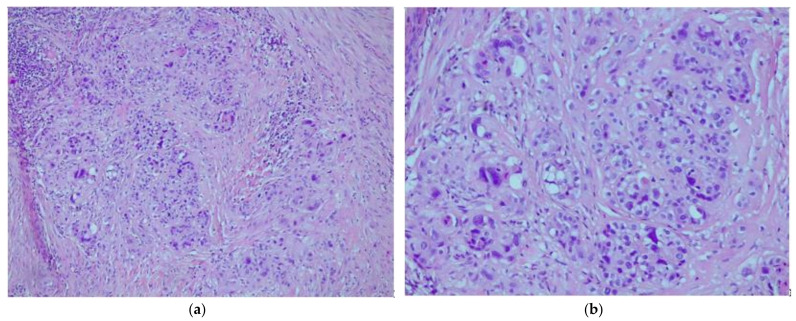
Histopathological exam of the sample (hematoxylin and eosin; H&E): (**a**) intermediate trophoblastic cells forming elongated structures and nests (H&E, 10×); (**b**) groups of the intermediate trophoblastic cells (H&E, 20×).

**Figure 2 medicina-58-00034-f002:**
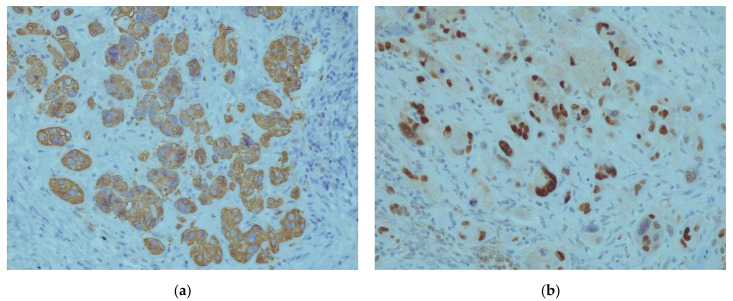
Immunohistochemical exam of the sample: (**a**) Diffuse strong positivity for CK AE1/AE3; (**b**) diffuse positivity for Cyclin E.

**Figure 3 medicina-58-00034-f003:**
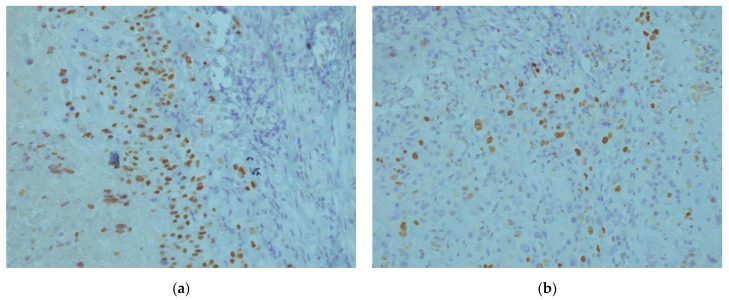
Immunohistochemical exam: (**a**) The sample was p63 diffusely positive; (**b**) Ki-67 nuclear labeling index of 20%.

**Figure 4 medicina-58-00034-f004:**
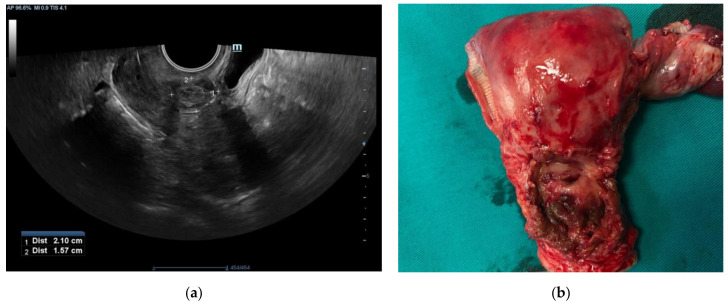
(**a**) Ultrasonographic exam revealing tumefaction of sharp borders to the surrounding tissue, with hypoechogenic halo; (**b**) a specimen of the uterus after the hysterectomy with the dehiscence in the region of previous Caesrean scar.

## Data Availability

Not applicable.

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
