# Peer review of "A Rare Case of Epithelioid Trophoblastic Tumor Presenting as Hematoma of a Caesarean Scar in the Lower Uterine Segment"

_medicina, 2021, doi:10.3390/medicina58010034_

Round 1
Reviewer 1 Report
The authors present a rare case of epithelioid trophoblastic tumor presenting as hematoma of Caesarean scar in the lower uterine segment. The title accurately reflects the case. The case involves an important area of health. The manuscript is well written in terms of clarity, style and has a logical construction. The discussion section explains the case in the context of published information. The conclusions accurately and clearly explain the main clinical message. The figures are of good quality and relevant to the clinical message. The references are appropriate and current.
However there are two issues that need correction:
- The figures and figure legends are mixed up. For example, figure legends in figure 1 describe immunohistochemical markers when they are hematoxylin and eosin photos.
- In the case report section, the authors describe the β-hCG as negative. They should describe it as not elevated or within normal values.
Author Response
Thank You for the analysis of our manuscript. We are grateful that You took Your time to review our scientific paper. We hope that, after the chages we made after Your remarks, our paper could now be considered for publishing in the journal.
Point 1: The figures and figure legends are mixed up. For example, figure legends in figure 1 describe immunohistochemical markers when they are hematoxylin and eosin photos.
Response 1: The figures have now been updated to reflect the mentions in the text. The legend of Figure 1 is now updated with the correct text. We have uploaded Figure 2 which shows the diffuse CK AE1/AE3 and Cycline E immunohistochemical positivity of the sample.
Point 2: In the case report section, the authors describe the β-hCG as negative. They should describe it as not elevated or within normal values
Response 2: β-hCG is now described as „not elevated“ or „within normal values“ everywhere in the manuscript.
Reviewer 2 Report
Dear Editors.
I would like to thank you for the opportunity to review the paper: “
A rare case of epithelioid trophoblastic tumor presenting as 2 hematoma of Caesarean scar in the lower uterine segment” submitted to the Medicina. The objective of the paper was to present a case report of ETT in cesarian scar.
Similarly, I would like to congratulate the authors for their interest in the topic.
I would like to highlight some aspects that deserve more reflection.
- Placental site trophoblastic tumor is also derived from chorionic type intermediate trophoblastic cells, as ETT. This sentence should be changed: “In contrast to other types of GTN (choriocarcinoma, invasive mole, and placental site trophoblastic tumor - PSTT) ETT is derived from chorionic type intermediate trophoblastic cells, forming nests and solid masses with nodular and expansive growth [5].”
- It would be possible to describe which test was used to measure the level of hCG, since the result was negative. Although with lower levels of hCG, the results are not usually negative in cases of ETT.
- In a quick search just in the years 2021 and 2020, I found 4 case reports similar to yours: ETT in caesarean scar. Please, remove this sentence: “To the best of our knowledge, this is only the second reported case of ETT existing in 82 the region of the Caesarean scar on the uterus.”
Emily Han-Chung Hsiue 1, Chiun Hsu, Li-Hui Tseng, Tzu-Pin Lu, Kuan-Ting Kuo. Epithelioid Trophoblastic Tumor Around an Abdominal Cesarean Scar: A Pathologic and Molecular Genetic Analysis. Int J Gynecol Pathol. 2017 Nov;36(6):562-567. doi: 10.1097/PGP.0000000000000366.
Kristin A.BlackaKristenSimoneaCassandraHirt-WalshabJeanelleSabourinac. Epithelioid trophoblastic tumor presenting as a Caesarean scar defect: A case report. Gynecologic Oncology Reports Volume 36, May 2021, 100715
Mardi Kavita, Sharma Shikha. Atypical postcesarean epithelioid trophoblastic lesion with cyst formation: A rare case report with review of literature
Clinical Cancer Investigation Journal Year : 2021 | Volume: 10 | Issue Number: 4 | Page: 219-222
- Black, Kri Simone, +1 authorJ. Sabourin. Epithelioid trophoblastic tumor presenting as a
Caesarean scar defect: A case report. Published 30 January 2021. Gynecologic Oncology Reports DOI:10.1016/j.gore.2021.100715
Author Response
We would like the Reviewer for the thoughtful comments and efforts towards improving our manuscript.
Once again, thank You for the analysis of our manuscript. We hope that our paper could now be considered for publishing in the journal.
Point 1: Placental site trophoblastic tumor is also derived from chorionic type intermediate trophoblastic cells, as ETT. This sentence should be changed: “In contrast to other types of GTN (choriocarcinoma, invasive mole, and placental site trophoblastic tumor - PSTT) ETT is derived from chorionic type intermediate trophoblastic cells, forming nests and solid masses with nodular and expansive growth [5].”
Response 1: We have changed the sentence which now follows as „ETT is derived from chorionic type intermediate trophoblastic cells, forming nests and solid masses with nodular and expansive growth”.
Point 2: It would be possible to describe which test was used to measure the level of hCG, since the result was negative. Although with lower levels of hCG, the results are not usually negative in cases of ETT.
Response 2: Our Clinic uses the chemiluminescent microparticle immunoassay (CMIA) method (Alinity, Abbot Ireland, Diagnostics Division Lisnamuck, Longford Co. Longford Ireland)
for quantitative and qualitative determination of beta-human chorionic gonadotropin (β-hCG) in human serum and plasma, where the detection limit for β-hCG is 2.3 mIU/mL and normal values ranges to 5 mIU/mL. Also, according to the remarks from Reviewer 1, we have replaced „negative“ β-hCG with „normal“ or „within normal values“ everywhere in the text.
Point 3: In a quick search just in the years 2021 and 2020, I found 4 case reports similar to yours: ETT in caesarean scar. Please, remove this sentence: “To the best of our knowledge, this is only the second reported case of ETT existing in 82 the region of the Caesarean scar on the uterus.”
Emily Han-Chung Hsiue 1, Chiun Hsu, Li-Hui Tseng, Tzu-Pin Lu, Kuan-Ting Kuo. Epithelioid Trophoblastic Tumor Around an Abdominal Cesarean Scar: A Pathologic and Molecular Genetic Analysis. Int J Gynecol Pathol. 2017 Nov;36(6):562-567. doi: 10.1097/PGP.0000000000000366.
Kristin A.BlackaKristenSimoneaCassandraHirt-WalshabJeanelleSabourinac. Epithelioid trophoblastic tumor presenting as a Caesarean scar defect: A case report. Gynecologic Oncology Reports Volume 36, May 2021, 100715
Mardi Kavita, Sharma Shikha. Atypical postcesarean epithelioid trophoblastic lesion with cyst formation: A rare case report with review of literature
Clinical Cancer Investigation Journal Year : 2021 | Volume: 10 | Issue Number: 4 | Page: 219-222
Black, Kri Simone, +1 authorJ. Sabourin. Epithelioid trophoblastic tumor presenting as a
Caesarean scar defect: A case report. Published 30 January 2021. Gynecologic Oncology Reports DOI:10.1016/j.gore.2021.100715
Response 2: Thank You for this remark. We have deleted the sentence and replaced it with „We found only a few reports of ETT presenting as a mass in the Caesarean scar, with two being synchronous with choriocarcinoma” and added references that You have mentioned.